# Bayesian analyses of radiocarbon dates suggest multiple origins of ceramic technology in Early Holocene Africa

Rocco Rotunno [1,2] ✉ & Enrico R. Crema [1,3]

Ceramic technology emerged and spread in Saharan Africa between the end of the 11th and the beginning of the 10th millennium cal BP during the so-called African Humid Period. This innovation is linked to hunter-gatherer-fisher groups adapting to changing and increased ecological productivity. Several putative points of origin and the resulting corridors of diffusion of this technology have been suggested in the literature, but there is currently no consensus on whether ceramics in this region originated as a single or multiple independent episodes of innovation. Here, we synthesise the available radiocarbon evidence associated with the presence and absence of ceramic technology in Early Holocene Africa and statistically model spatio-temporal diffusion processes using different combinations of putative origin points. The result of our model comparison provides support for either a dual or triple-origin model, with core areas potentially in the Central Sahara, Nile Valley and West Africa. These findings refine current debates on early pottery innovation, highlighting the role of localized technological choices, environmental factors and interregional interactions in shaping its spread.

The origin of pottery production marks a critical stage in the technological and cultural evolution of prehistoric societies. Recent research has demonstrated that the first ceramics, defined as multifunctional vessels, appeared independently in various regions across the globe and were initially associated with hunter-gatherer groups, preceding the emergence of productive economies[1]. Pottery first appeared between 20,000 and 12,000 calibrated years before present (cal BP) in East Asia, as evidenced by dated sequences at Xianrendong Cave, China[2]. Similar early pottery has been documented elsewhere in China[3], Russia[e.g4,5], and Japan[e.g6] between 17,000 and 15,000 cal BP.

In Africa, the emergence of pottery is situated around the end of the 12th millennium cal BP. While the current state of research does not definitively determine whether pottery was invented in a single or multiple regions in Africa, it is certain that pottery technology rapidly spread across a zone spanning approximately 2000 to 3000 km, extending through the central and southern Sahara, the northern Sahel, and along the Nile from the First Cataract down to the confluence of the White and Blue Nile (Fig. 1).

The earliest dates for pottery in the Central Sahara come from the sites of Tagalagal and Adrar Bous 10. At Tagalagal, an open-air site in the southern Aïr Massif, charcoal associated with pottery dates to around 10,700 cal BP ($9370 \pm 130$ BP, 172 Orsay), supported by thermoluminescence dates from two sherds, which range between $10,180 \pm 780$ and $9820 \pm 780$ BP[7,8]. At Adrar Bous 10, located outside the Aïr Massif on the fringes of the Tenere desert, the oldest pottery dates to around 10,300 cal BP[9]. In the nearby Temet site, lacustrine deposits dated to 10,900 cal BP (Paris-sud $9550 \pm 100$ BP)[10] provide indirect evidence of pottery production through a toothed object on a chlorite schist plaque, interpreted as a potter's comb. However, no direct evidence of pottery has been found at Temet, leaving the association of this site to the emergence of ceramic technology still uncertain.

[1]McDonald Institute for Archaeological Research, University of Cambridge, Cambridge, UK. [2]The Archaeological Mission in the Sahara, Sapienza University of Rome, Rome, Italy. [3]Department of Archaeology, University of Cambridge, Cambridge, UK. ✉e-mail: rotunno.rocco@gmail.com

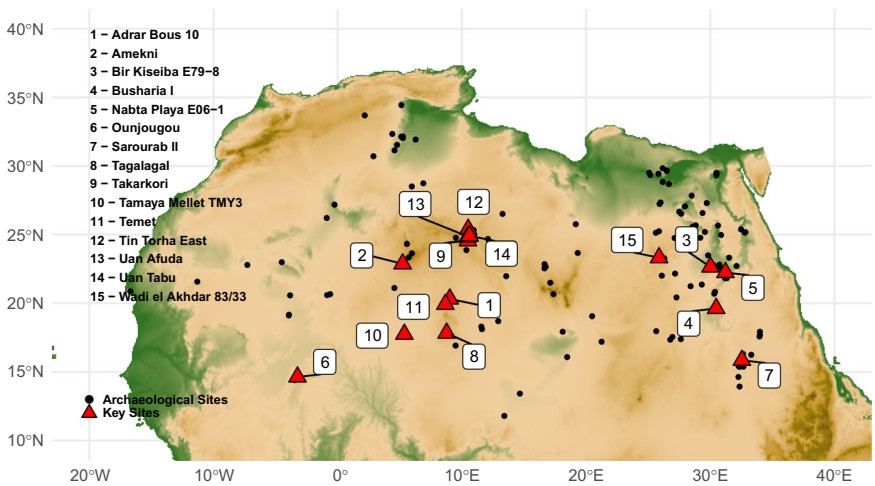

**Fig. 1 | Map of early pottery sites in Saharan Africa.** Distribution map of sample archaeological sites used in the analyses and locations of key sites mentioned in the text.

At Tamaya Mellet, another site in Niger, a faunal sample yielded a date of 9350 ± 170 BP (Gif-1728)[11,12], but human activity continued there until around 5000 BP. Early-type pottery with plant temper was discovered at this site, with two sherds dated to the 8th millennium BP (7550 ± 150 BP, Pa-1571, and 7415 ± 150 BP, Pa-1574)[12]. Notably, similarities between this pottery and Early Khartoum styles[13,14] raise the possibility of earlier pottery production, possibly as early as the 10th millennium BP, supported by both radiocarbon dates and comparisons with pottery from the Nile Valley[15,16] and the HA2 formation in Ounjougou.

In the Central Saharan massifs of southern Algeria and Libya, early pottery evidence has been reported since the very beginning of systematic archaeological research in the area in the past century. Stratigraphic evidence and radiocarbon dating suggest the presence of pottery by the 11th millennium cal BP at sites like Tin-Torha in the Tadrart Acacus, and Amekni in Algeria[17,18]. These early findings have since been reinforced by additional research, confirming ceramic production in the region from at least 10,200 cal BP[19,20]. These ceramics display a wide variety of decorative techniques but share consistent technological features, suggesting that more ancient centres of pottery production may still be undiscovered.

In the Bir Kiseiba region, alongside Nabta Playa, one of the most extensive ceramic chronologies in North-east Africa has been established, crucial for understanding prehistoric human occupation and cultural development[21,22]. This sequence, based on finds from numerous sites and extensive radiocarbon dating, spans the Early to Middle Holocene. The earliest pottery phase, known as El Adam, dates to the Early Holocene and has been found at a limited number of sites. This pottery exhibits similarities with contemporaneous early ceramic traditions in other parts of the Sahara and northern Sahel[15,22–24]. Despite its scarcity, El Adam pottery—decorated with simple impressed and rocker-stamped motifs—has been found across several sites in the area. While initial doubts concerning the association of these ceramics with early El Adam settlements were raised due to the limited number of sherds in situ, recent excavations at Nabta Playa have provided additional chronological data that better define the timing and characteristics of this early ceramic production[25].

At Bir Kiseiba's site E-79-8, three pottery sherds were found in sandy sediments at shallow depths[26], with the deepest sherd potentially displaced by surface activity. Three additional surface finds were also documented. Radiocarbon dates from charcoal samples range from 9820 ± 380 BP to 8920 ± 130 BP, but large error margins and the lack of stratigraphic context complicate direct association with the pottery. Nonetheless, recent data from site E-06-01 strengthen the case for early ceramic production in the region,

with samples such as Poz-19184 (9210 ± 150 BP) and Poz-19181 (9160 ± 180 BP)[25] providing further support for the early presence of pottery during the Early Holocene.

The earliest evidence of ceramic production in Africa, however, comes from the Ounjougou site in Mali. Ceramic sherds associated with a small bifacial lithic industry were discovered in Unit HA1 at the site of Ravin de la Mouche, which emerged during the wet phase of the earliest Holocene. This evidence is framed by terminus ante quem dates from the overlying HA2 deposit, which is dated to 9758 ± 70 BP (ETH-28746) and 9510 ± 70 BP (ETH-31279)[24]. These findings provide crucial insights into the early development of ceramics in West Africa and their association with early Holocene environmental conditions.

Given these lines of evidence, the origins and diffusion of pottery production in Africa are still debated, with three regions associated with the earliest evidence of this technology. The first one is located within the large mountain massifs of the Central Sahara (CS), with early sites like Adrar Bous 10 and Tagalagal. The second is in the Eastern Sahara and the Nile Valley (ESNV), with sites such as Bir Kiseiba E-79-8, Sarurab, and Wadi el Akhdar. The third is related to the most ancient evidence found at the Ounjougou site, situated in the southern Sahara/ Northern Sahel (SS) boundary zone.

The three proposed areas of origin correspond to distinct models or scenarios regarding the adoption and diffusion of pottery technology[27]. The first model links the adoption of ceramics to the exploitation of aquatic resources and wild grains in the Nile region[28]. The second suggests an independent development within remnant populations inhabiting refuge zones, particularly in specific eco-niches such as the mountainous areas of the CS[15,16]. The third suggests that the origin and diffusion of ceramics occurred in connection with a putative area located in the southern Sahara or further south, such as the Turkana region[29,30]. These hypotheses are not mutually exclusive and could all be equally and simultaneously plausible; however, they require systematic testing.

While the current state of research does not definitively determine whether pottery was invented in a single or multiple areas in Africa, the prevailing view among numerous scholars leans toward the idea of multiregional origins[15,16,29]. What is certain, however, is that pottery technology rapidly dispersed across a ca. 4000 km longitudinal zone extending through the southern Sahara and northern Sahel.

Here, we test statistically which of the scenarios is more strongly supported by the available evidence using a Bayesian modelling approach. Identifying the timing and the geographic point(s) of origin of any prehistoric innovation represents a considerable quantitative challenge due to the limitations imposed by small and often biased

samples available. True origination dates are undoubtedly earlier than the earliest dated samples, and the geographic proximity between the true point of origin and the location of the earliest sample is deeply confounded by the sampling error and spatial variation in sampling intensity. It follows that the most sensible approach is not to focus on individual sites but on the overall spatial and temporal distribution of the earliest samples available to benefit from all the information available. The intuition, in this case, is to build the inference under the simple assumption that sites located further from the origin point and/ or with more recent dates are less likely to be associated with the particular technology compared to those closer in space and time to the origination point.

Origin points studies in archaeology have been primarily focused on centres of domestication plants and animals[e.g.31–33] due in gran part to the availability of genomic data, which provides an additional layer of information derived from reconstructed genetic histories and the spatiotemporal distribution of genetic mutations. Still, in some cases, the only available line of evidence is the spatial and temporal distribution of presence data. For example, Silva and colleagues[34] identified the origin point of rice domestication in Asia by comparing seven locations hypothesised in the archaeological literature. Their approach consisted of fitting quantile regression to model how the earliest dates associated with rice agriculture change as a function of distance from putative origin points and subsequently using information criteria to determine the location providing the highest out-of-sample predictive accuracy.

Here, we built upon the approach developed by Silva et al.[34] and examined a sample of 855 radiocarbon dates from 259 site locations in the Sahara and Sahel ecoregions in Africa. We introduced three key changes to suit our inferential needs (see *materials and methods*). First, to maximise the information available, we carried out both quantile regression models where the dependent variable is the sample dates associated with pottery and the predictor variable is the distance from the putative origin point and binomial models based on the presence/absence of the technology in each context, with both time and distance from the putative origin point, as well as an interaction between the two terms, as predictor variables. By directly modelling presence-absence, we could consider evidence of absence and not be dictated by the absence of evidence. Second, we developed an approach that formally acknowledges the possibility of multiple points of origin, each with its own different origin time and diffusion rate. This was a key addition in our binomial model necessary to address our research question and consider the possibility that pottery technology might have been introduced independently at different locations. Third, we explicitly modelled and considered the uncertainty associated with each radiocarbon date and, where possible, site-level stratigraphic relationships. While the objective of the quantile regression model was an assessment of the plausibility of a single point of origin, the focus of the binomial model was the comparison, via information criteria, of seven unique combinations of one or more putative origination events.

## Results

Both the results of the quantile (Supplementary Table 1, Supplementary Fig. 1) and the binomial models (Fig. 2, Supplementary Figs. 2–8, Table 1, Supplementary Data 1) reject the possibility of a single point of origin for the diffusion of ceramic technology. All three single-origin quantile regression models show posterior ranges of the slope coefficient that are close to or inclusive of 0 (Supplementary Tab. 1), indicating how distance from putative origin alone is a poor predictor of the presence of ceramic at a particular time.

The binomial model, which accounts for nearly twice the sample size and the possibility of multiple points of origin, further reinforces these findings. All single-origin models ($m_1$ - $m_3$) returned a ΔWAIC larger than 29 (Table 1), effectively suggesting no support for these models. Results of the multi-origin models ($m_4$–$m_7$) yielded smaller ΔWAIC values, but gave the strongest support to a triple-origin model ($m_7$), followed by a dual model with Bir Kiseiba and Ounjougou Ravin de la Mouche ($m_5$; ΔWAIC = 7.55), and effectively no support for the remaining dual models, inclusive of Adrar Bous (ΔWAIC = 10.75).

A closer inspection of the parameters of the triple origin model ($m_7$) suggests a more prominent role played by Ounjougou Ravin de la Mouche in the initial diffusion process. The mean posterior probability of the presence of ceramic (Fig. 2, Supplementary Fig. 2,

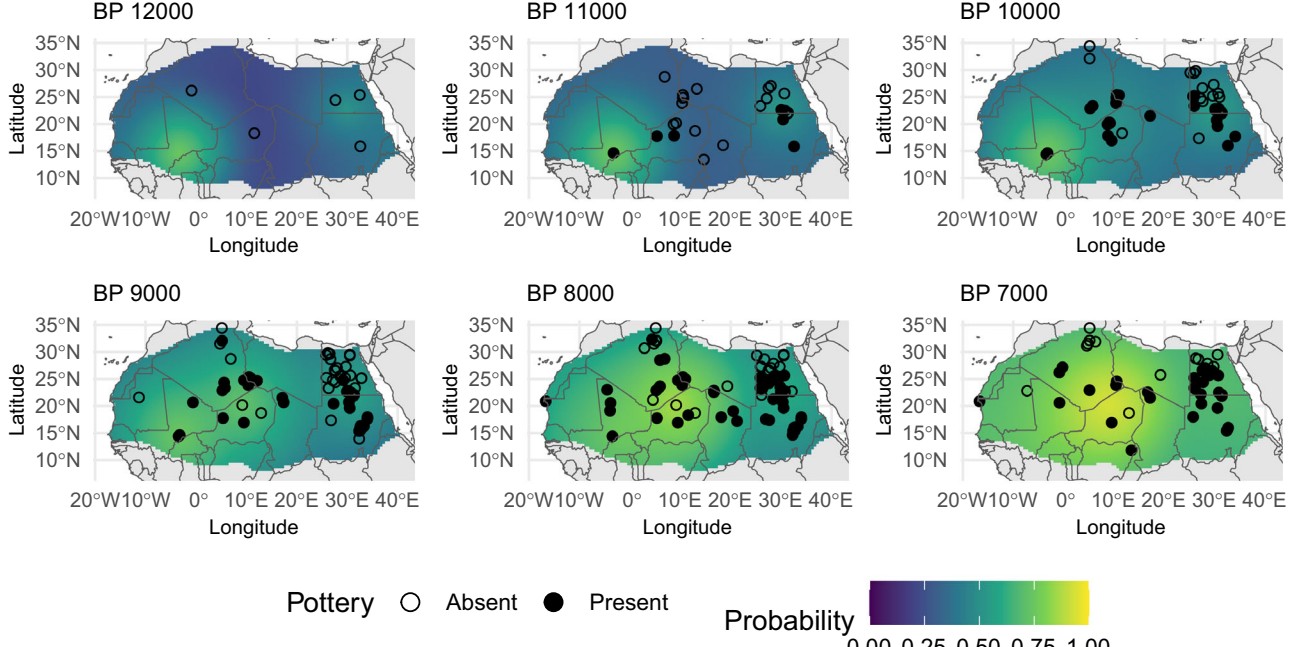

**Fig. 2 | Posterior mean probability for the presence of ceramic for model m7.** The filled points and circles show the locations of the samples with a calibrated cumulative probability mass over 0.5 for the interval spanning 500 years before and after each time slice.

Supplementary Data 1) shows clearly how western Africa reaches higher probabilities more rapidly compared to the rest of the window of analyses initially. The situation changes during later periods, from ca. 8000 cal BP when the initially slow diffusion from Adrar Bous saw an acceleration, suggested by the positive estimate of the interaction term. In contrast, the eastern origin point at Bir Kiseiba shows consistently lower probability of ceramic presence, most likely conditioned by the anisotropic diffusion process. Spatial analyses of the model residuals (Fig. 3) suggest indeed significant clustering of both low and high values in Egypt, indicating concentrations of sites where ceramics are predicted to be present but absent (e.g., northern Egypt at 8000 BP) as well as the other way around (e.g., Sudan, also at 8000 BP). While the small sample size does not allow more complex anisotropic models, these patterns might result from a slower rate of diffusion towards the north and a faster rate of diffusion south of Bir Kiseiba.

## Discussion

The results from the quantile and binomial models indicate that the diffusion of pottery technology in Africa did not follow a single-point origin model but rather a multiregional pattern. This multiregional model aligns with archaeological, environmental, palaeobiological and palaeolinguistic evidence, which suggests a complex history of population dynamics in North African during the Early Holocene[19,35–40] and points to overlapping zones of innovation across the Central and Eastern Sahara, the Sahel, and the Nile Valley. This supports the view that innovations such as pottery did not originate from a singular core area but rather emerged in multiple, potentially interconnected regions of innovation[15,16,27,29].

Ounjougou, where evidence from the Early Holocene points to the existence of pottery as early as 11,200 cal BP[24], exemplifies the southern Sahara-Sahel border area as one possible, but not exclusive, origin region for ceramic production. Contexts associated with similarly early dates suggest a chronological gradient of ceramic diffusion, moving progressively northward from the Sahel into more northern latitudes during the Early Holocene, coinciding with the peak of the African Humid Period. The regional context implies that ceramic technology was developed or adopted by local populations in response to ecological opportunities created by the northward expansion of the Intertropical Convergence Zone (ITCZ), which transformed arid landscapes into grasslands and wetlands, thereby supporting population growth and encouraging experimentation with innovative subsistence tools, including pottery. Complementary evidence from the lithic assemblage at Ounjougou, characterized by bifacial foliate arrowheads and other instruments, highlights the technological diversity and adaptability of these communities. The assortment in lithic technology reflects not only localized innovation but also broader cultural networks that likely facilitated the parallel diffusion of ceramics and lithic traditions. This underscores the interconnected nature of technological and cultural processes during the Early Holocene, as highlighted by comparison with other Saharan sites (e.g., Temet, Tagalagal, Adrar Bous10)[24,41]. These patterns suggest that the adoption of pottery occurred within a wider context of technological renewal; rather than emerging in isolation, early ceramics appear to have developed alongside changes in lithic production within distinct cultural settings. The recurrence of this association in various regions points to a shared dynamic in which technological shifts were shaped by local adaptations to environmental change and population reorganisation.

In CS, sites such as Adrar Bous 10 and Tagalalal in the Central Sahara Massifs and Sarurab 2, Bir Kiseiba E-79-8 and Wadi Al Akhdar in the Nile Valley and Eastern Sahara point possibly to various and more or less independent innovation centre though interrelated. In fact, to our present state of knowledge the evidence from Ounjougou may have resulted from an innovation/invention centre located in the

**Table 1 | WAIC-based model comparison for the binomial model (b: Bir Kiseiba; a: Adrar Bous 10; o: Ounjougou Ravin de la Mouche)**

| Model | Origin Point(s) | WAIC | ΔWAIC | Weight |
|---|---|---|---|---|
| $m_7$ | $b + a + o$ | 8336.11 | 0 | 0.973 |
| $m_5$ | $b + o$ | 8343.66 | 7.551 | 0.022 |
| $m_4$ | $b + a$ | 8346.87 | 10.756 | 0.004 |
| $m_6$ | $a + o$ | 8357.32 | 21.208 | 0 |
| $m_2$ | $a$ | 8365.55 | 29.436 | 0 |
| $m3$ | $o$ | 8366.29 | 30.176 | 0 |
| $m_1$ | $b$ | 8374.98 | 38.871 | 0 |

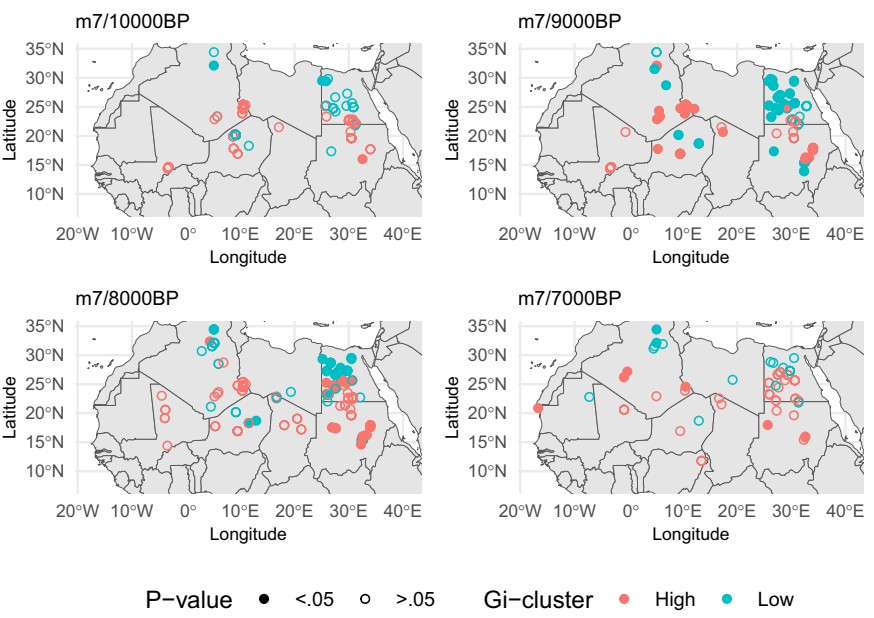

**Fig. 3 | Residuals for the multi-origin points model.** Getis-Ord Gi statistics with locations of significant clustering of positive (filled red points) and negative (filled blue points) residuals for selected time slices of model m7. Two-sided P-values were obtained using 999 permutations.

current Sahelo-Sudanian area from where it expanded following the greening of the Sahara, a population expansion that went at pace with the northward expansion of the ITCZ at the beginning of the Early Holocene[36,40,42–45], and expanded towards the north in the Central and Eastern Sahara. The sites with the most ancient dates from the former area, Tagalgal and Adrar Bous 10, show assemblages with already diversified and internal variability in decoration, style and production processes, symptoms possibly of the adoption of an already developed technology[16,27,46,47]. The same can be said for the Sudanese sites, like Sarourab 2 or Busharia 1[48–52].

The results from the binomial models further indicate that the diffusion process was not uniform across Africa. The more rapid spread of ceramics in some parts of the Sahara, particularly in regions north of Ounjougou, suggests that environmental or cultural factors may have facilitated quicker adoption of the technology in these areas. Conversely, the slower diffusion rates observed in very northern regions, such as Egypt north of the second and first cataract, may reflect environmental barriers or social factors that delayed the adoption of pottery[36]. The clustering of residuals in Egypt, where ceramics are predicted but absent, suggests that environmental constraints or cultural resistance could have played a role in slowing the diffusion process. These patterns of anisotropic diffusion highlight the complexity of technological transmission in prehistoric Africa and suggest that localized factors significantly influenced its spread[53–57].

The Eastern Sahara and parts of the Nile Valley, as aforementioned, also emerges as an important area for pottery adoption and diffusion. For instance, the archaeological record for the Bir Kiseiba and Nata Playa area, supported by radiocarbon dates from sites like E-06-1[25] and new dates from post-glacial reoccupation sites along the Nile, such as those reported by refs. 58, 59, emphasizes the Nile Valley's role as a significant corridor for the movement of people and technologies. The ceramic assemblages from these and other sites between the second and third Nile cataracts, display localized adaptations, such as variations in paste composition, temper materials, and decorative motifs[47,60–62], underscoring a regionalized mode of ceramic transmission. The recent finds and radiocarbon data from the Sphinx site at Jebel Sabaloka in central Sudan suggest a nuanced pattern of cultural diffusion and local adaptation. The site's earliest occupation, dated to 10.7–10.5 ka cal BP, demonstrates a synchronous postglacial reoccupation of the Sudanese Nile with regions of the Sahara, challenging previous assumptions of a significant delay in occupation south of the Second Nile Cataract[58,60]. The regional data also emphasize that diffusion processes were not isotropic. For instance, findings at Sphinx align with Early Khartoum cultural markers, including early pottery production alongside lithic and ostrich eggshell bead technologies, as part of an integrated set[58,60]. This reflects a technological continuity likely carried by groups dispersing from tropical African regions, facilitated by the ecological corridors of the Nile and adjacent areas. The ceramic assemblages at Bir Kiseiba and Nabta Playa further underscore the Nile Valley's importance as a corridor for the movement of people and technologies during this period. The technological variability in these assemblages, including paste composition, temper materials, and decorative motifs, indicates both regional adaptations and broader cultural linkages[22,63].

Similarly, the Central Sahara contributed to this diffusion process, as evidenced by the ceramic assemblages at Adrar Bous and Tagalgal and surrounding regions[46,47]. These sites, along with others in southern Libya and Algeria, exhibit early pottery production dating to as early as 10,650 cal BP. The pottery from these regions suggests the adoption of an already developed technology, possibly introduced through interactions with neighbouring regions where ceramic production was already existent. Assemblages from sites like Uan Afuda[64], Ti-n-Torha East[17,65], Takarkori[20], with pottery bearing layers dated to 10,200 cal BP, provide additional evidence of the integration of pottery into the subsistence strategies of complex hunter-gatherer-fisher communities[19,64,66,67]. These groups utilized pottery for processing and storing wild plants, aquatic resources, and reflecting a broad spectrum of economic practices[19,66,68,69]. Associated with a more sedentary lifestyle, this earliest pottery bearing phase (Late Acacus, 10,200–8000 cal BP), witnesses also a higher utilization of grinding tools, plant processing and complex forms of animal management[65,70–73], integrating fish and wild grasses, like wild sorghum and Pennisetum sp. into the resources exploitation strategies evidencing a high degree of ecological knowledge and specialization[73–75]. The archaeological synchronous presence in the record of different sites in the Tadrart Acacus might suggest an arrival or circulation of this technology rather than a fully local or independent emergence. The archaeometric study of this pottery suggests the possible extent of interactions and exchange among these communities. Raw materials used in the production process were sourced both locally and from more distant locations, including outcrops in the Tassili N'Ajjer region situated about 70 km away[76,77]. While the Central Sahara shows evidence of some of the earliest pottery traditions, the pattern of diffusion appears initially patchy and relatively slow, with early innovations concentrated in a few key areas. However, from around 9000–8000 cal BP onwards, the region undergoes a more marked and widespread integration of ceramic technology. This accelerated phase likely reflects intensifying regional connectivity, population movements, and evolving socio-economic strategies that promoted broader adoption. The shift from initial adoption to more rapid diffusion underscores the heterogeneous tempo of technological uptake across the Central Sahara, shaped by local conditions and broader cultural networks.

Understanding the mode and tempo of Early Holocene hunter-gatherer pottery dispersal into the Sahara and North Africa provides insights into the mechanisms responsible for the cultural transmission in these contexts. Despite intrinsic limitations such as the preservation biases of the archaeological record, the data reveal some discernible patterns. Most data are derived from palimpsest-like deposits, which do not always represent the earliest phase of pottery use in each locality, hence introducing some noise into the spatio-temporal models. While demic diffusion may have been a factor, the pace of dissemination suggests that pottery production was transmitted probably through knowledge transfer across established networks among dispersed hunter-gatherer communities and via a multicentric early emergence pattern. This can be further supported by widespread commonalities in regional distribution and formal attributes of decoration and style[16]. Pottery analysis highlights the use of mineral and coarse-tempered fabrics, and vegetal tempering, while differences in manufacturing techniques, particularly in paste recipes, indicate the use of local and sub-local raw material sources, supporting evidence for primarily local production[16]. Decorations, predominantly impressed, reveal regional and chronological variations but share overarching traits such as the prevalence of the rocker motion technique[15,50,78,79]. These patterns suggest local production within socio-cultural frameworks connected by complex interrelationships[28,29,80–82].

The widely debated diffusion of incised and dotted wavy line (IWL and DWL) decoration across North Africa exemplifies these networks[15,30,50,83]. However, attributing contacts or influences based solely on decoration is neither justifiable nor reliable, given the variability of this attribute. Communalities across other cultural realms and the geographical and chronological contexts suggest that these traits reflect intricate dynamics rather than pan-regional uniformity.

Early Holocene communities in North-East Africa adopted broad-spectrum subsistence strategies, economic intensification, and increasing sedentism as effective responses to climatic and environmental instability. Strategies such as delayed return, food storage, and pottery production underscore complex resource management systems[19,23,84]. In the El Nabta Basin, for instance, early evidence of wild plant exploitation and storage dates to the 10th–9th millennium cal

BP[66,85] with similar practices documented along the Nile and in other Saharan regions[84,86,87]. Economic specialization involved the intensive exploitation of predictable resources incorporated into delayed-use strategies. Storage methods, both physical and environmental[88–90], combined with a degree of sedentariness, characterize these populations. Numerous groups exhibited a reliance on aquatic resources, evidenced at sites ranging from the Nile shores to the Central Sahara[23,75,84]. Ethnographic parallels indicate that such groups often exhibit higher population densities, smaller territories, increased residential stability and greater technological specialization[91,92]. Although these traits were not uniformly shared across all groups, their variability across eco-niches and eco-tones suggests complex population dynamic,s though the extent to which environmental interactions shaped archaeological assemblage similarities or reflect past social networks and connectivity at a regional scale remains unclear[93,94].

The cultural trajectories of hunter-gatherer-fishers groups suggest that the introduction and adoption of pottery were closely associated with broader shifts in socio-economic organization, likely influenced by environmental and demographic changes during the early Holocene. Increased specialization in the exploitation of wet/humid environments, well documented through archaeozoological and archaeological evidence, appears as a recurrent observed pattern[75,95,96]. These developments correlate with multiple lines of evidence suggesting that the African Humid Period (AHP) facilitated regional population expansion and mobility within ecologically favourable zones[35,40]. Genetic and bioarchaeological data[39,97–100] point to episodes of demographic growth and interaction embedded in long-term population continuity, without substantial exogenous gene flow or demographic replacement. Ethnolinguistic studies[35,37,38,101] similarly indicate patterns of linguistic diversification reflecting regionally structured demographic and cultural processes. Combined with emerging archaeological evidence, these insights point to the complexity of early Holocene dynamics, structured by long-term historical trajectories and enduring patterns of interregional connectivity across Sub-Saharan and Sahelian regions[27,41,102–104].

Multiple hypotheses have been proposed regarding the origins, adoption, and initial functions of pottery[105]. Many scholars emphasise its role in food processing, particularly boiling and steaming, which increased digestibility and palatability for plant- and fish-based diets. Pottery vessels were likely used for stews or soups[28,106–109]. This marks a significant culinary transition in which pottery became an integral component of new food preparation and consumption habits, including the preparation of porridge-like meals[84,108–110]. The understanding that pottery's physical properties facilitated new food technologies highlights the functional benefits of cooking in ceramic vessels[1,5,68,105,111,112].

Beyond cooking, early pottery may have fulfilled additional roles related to food processing and preservation, though direct evidence from residue or use-wear analyses remains limited[60,64,68,69,71,95,113,114]. Ceramic vessels likely offered advantages over organic containers in terms of resistance to pests and rodents and may have facilitated new strategies for managing food resources. Over time, pottery may have been perceived as a cost-effective and efficient technology useful in response to subsistence needs. Compared to other container technologies, pottery offered distinct advantages in production efficiency and durability[1,114,115]. The transmission of pottery among Saharan hunter-gatherers was thus the outcome of extensive social interactions spanning wide geographic areas, consistent with a pattern of multiple, regionally distinct centres of ceramic innovation. Compared to other containment technologies, pottery was a relatively low-cost innovation, benefiting from the availability of raw materials and the communal nature of knowledge transfer within household and kinship structures[116,117]. Observations from the earliest pottery-associated sequence at Temet indicate that some containers were crafted from polished stone, rather than fired clay[27]. This finding suggests that skeuomorphic practices may have informed the inception and adoption of ceramic technologies[118]. A more conspicuous example of skeuomorphism is found in the decorative motifs of these early ceramics—such as rocker-stamped, packed dotted zigzags—that might appear to replicate basketry. Pottery, being more easily shaped than carved stone and more durable than basketry, thus preserves and transmits both the functional and aesthetic attributes of preceding container forms[105].

In societies with mobility patterns structured around seasonal resource availability, the adoption and diffusion of pottery may have been facilitated by contexts allowing repeated access to clays sources, fuel and other raw materials[105,111,119,120]. Without implying a causal relationship between reduced mobility and ceramic production, logistically organized settlement patterns, particularly during the African Humid Period, could have supported the regular practice and transmission of pottery-making. The expansion of lakes, rivers, and fertile landscapes likely encouraged longer seasonal stays in specific locales, creating conditions conducive to ceramic innovation. Given the likelihood of multiple, independent origins of pottery in Africa, as suggested by the results presented here, understanding ceramic adoption and use requires attention to the specific ecological and social settings in which early ceramics emerged.

In conclusion, our statistical analyses suggest that rather than originating from a single innovation centre, multiple regions contributed to the broader adoption of ceramics across the continent. These findings underscore the necessity of ongoing archaeological research to brighten the intricate cultural and technological processes that shaped prehistoric African societies, underpinning the socio-economic developments of Early Holocene Africa.

## Methods

### Radiocarbon Database and Sample Preparation

The database comprises 855 radiocarbon dates from 259 sites, where the presence or absence of pottery has been checked through a thorough review of recovery contexts described in the available literature. When available, we derived the presumed temporal relationship between samples from the same site based on stratigraphic information of the contexts of recovery. From an initial pool of approximately 3200 radiocarbon dates for North-East Africa, as documented in various repositories, we have systematically selected a subset based on our defined criteria. The selection process involved a comparative analysis of existing radiocarbon repositories, with particular emphasis on dates within the 12,000 to 7000 BP range. Geographically, the study encompasses the Sahara and Sahel eco-regions. The analysis deliberately excludes sites from Mediterranean Africa due to the distinct cultural dynamics related to pottery adoption in connection with the Neolithic expansion in the Mediterranean[121–124]. Consequently, the study area is confined to a latitude range from 34°N in the north to 10°N in the south.

To control for the potential bias introduced by inter-site variation in sampling intensity, we first followed the same protocol used in the demographic inference on radiocarbon dates[125] by 'binning' samples that were sufficiently close in time within the same site using the *binPrep* function in the *rcarbon* R package[126] with a time distance threshold of 100 years. We then selected randomly but prioritised samples with the lowest measurement error, one representative date for each bin. Our final dataset consisted of 586 radiocarbon dates.

### Bayesian quantile regression

We first modelled the dispersal of ceramic technology examining the relationship between radiocarbon dates associated with the presence of pottery ($n = 337$) and their great arc distance from three putative origin locations: (a) Bir Kiseiba (30.02 E, 22.65 N; Central South Nile), (b) Adrar Bous (9.03 E, 20.32 N; Central Saharan Massifs), and c)

Ounjougou Ravin de la Mouche (3.23 W, 14.63 N; South-Western Sahara). We followed the same procedure described in (33), where the calendar date $\theta_i$ of the sample $i$ was modelled using an asymmetric Laplace distribution[127]:

$$\theta_i \sim \text{AsymLaplace}(\mu_i, \lambda, \tau) \qquad (1)$$

where $\mu_i$ is the location parameter of the sample $i$, $\lambda$ is the scale parameter, and $\tau$ is the quantile of interest, in this case, the 5th percentile. The location parameter $\mu_i$ was defined using the following linear expression:

$$\mu_i = \gamma_0 + \gamma_1 di \qquad (2)$$

where $\gamma_0$ and $\gamma_1$ are the intercept and the slope parameters and $d_i$ is the distance between the putative origin and the location of the sample $i$. Finally, the relationship between the calendar date $\theta_i$ and the $^{14}$C age $x_i$ was modelled as follows:

$$x_i \sim \text{Normal}(f(\theta_i), \sigma_i) \qquad (3)$$

where $f(\theta_i)$ is the corresponding $^{14}$C age to the calendar date $\theta$ on the calibration curve and $\sigma_i$ and is the square root of the sum of the squares of the measurement error associated to each radiocarbon date and the error on the calibration curve at $\theta$.

### Single and Multi-origin Bayesian Binomial Model

We constructed and fitted Bayesian Binomial models where the probability $p_i$ of a sample being associated with ceramic is a function of its calendar date $\theta_i$ and its distance to one or more of the three origin locations. We fitted seven different models (see Tab. 1), three with a single origin ($m_1$ - $m_3$), and four with varying combinations of two ($m_4$ - $m_6$) or three origins ($m_7$). In the case of a single origin, our model is essentially a binomial GLM:

$$y_i \sim \text{Bernoulli}(p_i) \qquad (4)$$

$$\text{logit}(p_i) = \beta_0 - \beta_{time}\theta_i - \beta_{distance}d_i + \beta_{interaction}d_i\theta_i \qquad (5)$$

where $y_i$ is the observed binary response (1=presence, 0=absence of ceramic), $\beta_0, \beta_{time}, \beta_{distance}$, and $\beta_{interaction}$ are model parameters, and $\theta_i$ and $d_i$ are the calendar date and distance from origin for the sample $i$. The interaction term $\beta_{interaction}$ term provides flexibility to the model to account for diffusion processes that can accelerate or decelerate at larger distances from the origin or later time periods. As for the quantile model described above, the relationship between $\theta_i$ and the $^{14}$C age of the sample is modelled by equation [3]. In the case of models $m_4$ - $m_7$, we applied equation [5] for each putative point of origin separately and then selected the highest $p_i$ for equation [4]. We compared our seven models using WAIC[128]. We have also examined the spatial autocorrelation of model residuals (calculated as the difference between $y_i$ and the posterior mean of $p_i$) using local Getis-Ord G statistic[129] for selected time-windows (i.e., considering samples with cumulative calibrated probability equal or above 0.5 between the start and end date of the time-window).

### Implementation

All analyses were performed using R statistical computing language[130]. Bayesian analyses were designed and fitted using Nimble probabilistic programming language via the Nimble[131] and nimbleCarbon R packages. Weakly informative priors were selected for all models (see SI text for detailed model definition), and in the binomial model, all variables were scaled and centred to aid the fitting process. All models were fitted using four chains, with 100,000 iterations for the quantile regression models and 500,000 for the binomial models. For both model types, we discarded half of the iterations for burn-in and thinned the samples to retain 10,000 posterior values for each chain. Model convergence was evaluated using the Gelman-Rubin statistic[132] and Effective Sample Size. Local measures of spatial-autocorrelation (Getis-Ord G statistic[129]) of the model residuals were computed using the *sfdep* R package[133].

### Reporting summary

Further information on research design is available in the Nature Portfolio Reporting Summary linked to this article.

## Data availability

All data necessary to reproduce the analyses are available on the GitHub repository https://github.com/roccorot/OriginPotteryNAfrica, and archived in the following Zenodo repository: https://doi.org/10.5281/zenodo.16895187.

## Code availability

All code necessary to reproduce the analyses are available on the GitHub repository https://github.com/roccorot/OriginPotteryNAfrica, and archived in the following Zenodo repository: https://doi.org/10.5281/zenodo.16895187.

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

## Acknowledgements

This work contributes to the project EHSCAN-Exploring Early Holocene Saharan Cultural Adaptation and social Networks through socio-ecological inferential modelling, supported by the UKRI Postdoctoral Fellowship Guarantee Grant (UKRI Funded/MSCA Actions Awarded), Grant Reference: EP/Y028430/1. We would like to thank the McDonald Institute for Archaeological Research, University of Cambridge, for providing institutional support and a stimulating research environment that greatly contributed to this work.

## Author contributions

Conceptualization: E.R.C. and R.R. Methodology: E.R.C. and R.R. Data Collection: R.R Statistical analyses: E.R.C. and R.R Writing-original draft: E.R.C. and R.R. Writing-review and editing: E.R.C. and R.R.

## Competing interests

The authors declare no competing interests.
