## [Transparent Peer Review file · Nature Communications]

Bayesian analyses of radiocarbon dates suggest multiple origins of ceramic technology in Early Holocene Africa

Corresponding Author: Dr Rocco Rotunno

Version 0:

Reviewer comments:

Reviewer #1

(Remarks to the Author)

Review of NCOMMS-25-15778 - Bayesian analyses of radiocarbon dates suggest multiple origins of ceramic technology in Early Holocene Africa

Thanks for allowing me to review this manuscript on the origins and diffusion of ceramic technology in Africa. As I will outline below, I really like this study. It addresses an important question in prehistory, provides a logical and innovative methodological approach to the question, and thus meets all criteria for publication in Nature Communications. I provide comments below that do not undermine the authors' arguments. Rather, my questions and suggestions regard the nature of the cultural diffusion mechanism assumed by the statistical models. Mainly, the models' ability to detect a broader spectrum of spatiotemporal diffusion signals. I believe that the authors will easily address my questions and comments. Consequently, I highly recommend this manuscript for publication in this journal.

Contextual Importance of the Study

For context, I would like to highlight that the textbook I use for my introductory archaeology course, *The Human Past: World Prehistory and the Development of Human Societies* (July 2024, 5th edition, Thames & Hudson), presents the spread of ceramics in Africa as a seemingly diffusion process from a single origin. Specifically, it states:

"From an open-air site in Mali called Ounjougou, at the southern edge of the present Sahara, comes the earliest pottery, which dates from before 9400 BCE. This is the oldest pottery found in Africa and predates that from Southwest Asia... From here, at the end of the tenth century to the beginning of the ninth century BCE, ceramics spread into other parts of Africa as the desert zone became increasingly greener." (p. 327)

To me, this framing reflects a textbook single-origin diffusionist model. It posits pottery as emerging from a single center and radiating outward in response to environmental changes.

The study is particularly significant because it directly tests this single-origin diffusion hypothesis using advanced Bayesian modeling. The study's findings, which favor multiple independent origins of pottery in Africa, challenge the single-origin diffusion model taught in classrooms (at least by the textbook I use in mine). Consequently, this paper can potentially reshape how the evolution of early African technology (e.g., ceramic traditions) is taught to future archaeologists and students more broadly.

To do this, the study combines Bayesian quantile regression and Bayesian binomial regression, along with local spatial autocorrelation analyses (I know these as "LISAs" following Anselin's work). The quantile regression framework focuses on estimating the early temporal quantile of the dataset (5th percentile), effectively isolating the earliest archaeological occurrences of pottery for analysis. One positive point to note about this approach, which others may miss, is that, like OLS, quantile regression uses all data points to estimate effects at a given quantile. Even though the study targets a specific quantile, the whole dataset contributes to the estimation. This is especially helpful in the distribution's tails, where individual quantile points are sparse. The method thus "borrows strength" in areas with potentially little data, leveraging information from the overall data structure. This strengthens the signal detection ability of their model (s).

The authors then supplement this with binomial models to estimate the probability of pottery presence across space and time. These models allow them to compare competing hypotheses of ceramic spread based on proposed origin locations and temporal dynamics.

Their results show that none of the tested centers acted as a sole origin of pottery spread, as might be expected under a classical single-origin diffusion model. Instead, the best-performing models consistently favored scenarios involving three widely geographically dispersed innovation centers. These results suggest independent emergence of ceramic technology across the continent. Their quantile regression and binomial modeling also indicate positive time effects but surprisingly small distance effects. The latter, to the extent of being zero. The probability of pottery presence does not covary with increasing distance from the proposed origins as strongly as expected from a single-origin diffusion model.

Thus, they reject the hypothesis of a single origin for the spread of pottery in Africa. These results challenge assumptions of a simple diffusion process and highlight the complexity of early African ceramic traditions.

This is an innovative, well-conducted study, and the authors should be commended for their work. First, the Bayesian approach is novel. The quantile regression ensures that only the earliest subset of radiocarbon dates is evaluated for the origins of pottery amongst themselves. Then, they use binomial Bayesian regression models to supplement their analysis with the larger sample.

I have a few suggestions regarding the structure of their models. I fully acknowledge that my thinking may be imperfect. Therefore, I will outline my reasoning openly below.

The authors are welcome to correct my reasoning if they think I am wrong. If they do so, this would naturally call my resulting suggestions into question, and they should act accordingly.

Essentially, the study's models are limited to main effects only. The covariates are standardized space (distance) and time (C14). The key is that each variable is modeled assuming its effect does not vary across the range of the other variable. In particular, the influence of space is presumed constant across time values, and vice versa, rather than allowing for the possibility that their relationship might change over time or across spatial gradients. I strongly encourage the authors to incorporate an interaction term in their models. From the perspective of the diffusion process itself, there is good reason to expect that the rate of spatial spread does not remain constant over time. To clarify my reasoning, let me first explain how I conceptualize diffusion and why I recommend exploring this interaction effect.

I apologize if what follows seems overly detailed and lacking in citations. The authors are eminent figures in quantitatively modeling cultural transmission and ceramics, particularly in archaeology, and I deeply admire their work. Given their familiarity with diffusion models such as Wright-Fisher and MacArthur-Wilson, I will avoid belaboring well-known points but will lay out some basic thoughts for context.

My intention here is simply to explain my rationale. I believe this is a relatively straightforward and worthwhile extension that could benefit the study.

Expectations for Diffusion Data and Statistical Modeling

As given, the central pattern in diffusion processes is progressive expansion across space and time, whether of technologies like pottery, cultural innovations, or diseases. Diffusion typically begins at a localized origin point and radiates outward as time advances. In an additive model, as the study currently presents, the effect of space is treated as constant over time. This means that the spatial coefficient estimate reflects only an average influence across the full temporal range of the data. From the first principles of transmission processes, this spatial effect may appear small early on because adoption is mainly localized and the overall rate of spread is slow. However, as time progresses, initial adopters become secondary transmitters. As populations grow and interact, the influence of space amplifies, accelerating the spread across broader regions.

An additive model cannot capture this dynamic compounding effect, because it assumes spatial and temporal influences operate independently. In such cases, the estimated effect of space might appear small at first glance, which risks leading to misinterpretation. By contrast, including an interaction term between space and time allows the model to represent and test how the effect of space changes over time (and vice versa). The interaction term effectively increases the effect of space as the network of transmitters grows. Without this interaction, a model risks underestimating and failing to test the role of space. It might misinterpret a seemingly minor spatial effect as negligible, when in fact it can become a key driver of accelerating diffusion.

In this context, even modest-sized interaction estimates can exert substantial influence, particularly in models with standardized predictors and logit transformations. In these models, small shifts in parameter estimates can dramatically alter predicted probabilities.

Retooling the Bayesian Quantile Regression Models

Thanks to the code and data provided, I restructured the original Bayesian quantile regression framework to explore this further. Specifically, I included an interaction term between time and distance in their nimble quantile model code (adding: `beta_interaction * distance[i] * time[i]`; and modeling it as `beta_interaction ~ dnorm(0, sd = 1)`). This addition enables a direct test of whether the effect of distance on pottery presence changes over time.

After running the modified structure, using their "e" model, I examined each parameter's 95% Highest Posterior Density Interval (HPDI). The intercept (alpha) remained around 11,000 BP, as expected. The time and distance coefficients spanned negative to positive ranges, reflecting considerable uncertainty. Most importantly, the interaction term's 95% HPDI ranged from approximately -0.40 to $+0.94$. Although this interval includes zero, approximately 75% of the posterior distribution lies above zero. While not conclusive, this posterior skew provides moderate Bayesian support for a positive interaction effect. In other words, while distance initially suppresses the probability of pottery presence, this inhibitory effect weakens over time. This pattern aligns with expectations from diffusion theory.

I acknowledge that my reanalysis was run with fewer MCMC iterations than the original authors used. For practical reasons, I ran only a few thousand iterations compared to their 100,000 (for the quantreg) and 500,000 (for the binomial). Given that the authors are deeply familiar with their own framework and likely have optimized workflows, I strongly recommend that they run dedicated interaction models themselves to explore this hypothesis fully. They are undoubtedly better positioned to execute these models with the rigor and convergence diagnostics necessary to draw robust conclusions.

Model "e," even in my cursory evaluation, may already reveal a more nuanced spatiotemporal dynamic. Over time, distance seems to become less of a barrier to the spread of pottery. This result aligns with the core mechanisms of cultural diffusion. With a more thorough analysis, the authors can examine this pattern more deeply and evaluate its robustness.

Re-estimating the Binomial Diffusion Models with Interaction

I also re-ran the original binomial models to test space-time dynamics further. I introduced an interaction term, β_3 ($\beta_3 \sim \text{T}(\text{dnorm}(\text{mean}=0, \text{sd}=0.5), 0, \text{Inf})$), between time and distance. This modification explicitly tests whether the effect of distance changes over time.

Comparing models using WAIC, I observed the following:

- Model m5 (e+o origins) showed a slight WAIC improvement (-1.22), suggesting the interaction modestly improves model fit.
- Model m7 (e+a+o origins) showed an increased WAIC ($+14.79$), indicating that the interaction does not benefit the fit and may overcomplicate the model.
- Other models showed minor changes, generally within the range of noise. But again, this may be due to my relatively low number of MCMC.

As noted earlier, my reruns were performed at lower iteration counts than the original authors' thorough MCMC processes. While these preliminary results suggest that the interaction term improves some models, especially those with fewer origins, I do not believe they radically change the overall model landscape. Nevertheless, the authors' indication that the spatial effect is nil merits further testing, particularly the interaction term. As I have said above, given their familiarity with the data and modeling framework, they could better investigate the role of interaction terms. Running longer chains and complete diagnostics could provide a fuller understanding.

My limited modeling of the interaction term seems to have improved models with fewer origins, like model m5, but had limited impact across the broader model set. This pattern suggests that while time and space are not wholly independent, the strength of their interaction varies depending on the assumed number and placement of origins.

To clarify, my exercise cannot replace my recommendation that they perform a more thorough examination of my preliminary analysis. I would argue that what I did is better than providing a verbal model or thought experiment, but it remains an initial exploration. Nonetheless, I (perhaps naively) believe it highlights the potential for uncovering further dynamics within the data. Given the structure of the diffusion process and the indications from my exploratory quantile regression and binomial reanalyses, I believe this direction warrants serious attention.

Code Comments and Reproducibility Notes

While revising the code, I encountered a few minor issues:

1. The `here()` package requires proper project root recognition. For reproducibility, it might help to set the working directory explicitly or use an RStudio Project to manage paths. Maybe because I am old school (it hurts a little to say that!), I replaced `load(here('data', 'input_binom.RData'))` with `load("../data/input_binom.RData")`.
2. The function `select()` was ambiguous in my environment due to conflicts with other loaded packages. I imagine others may have a similar problem. Explicitly using `dplyr::select()` resolved this.
3. I spent considerable time understanding the loop structures handling stratigraphic constraints. Some additional inline comments in the future would enhance clarity and ease replication.

These adjustments improve robustness and reproducibility across environments.

Summary

In conclusion, the paper under evaluation offers a methodologically rigorous and empirically grounded challenge to single-origin diffusion hypotheses of African ceramics. By employing Bayesian frameworks, the authors provide evidence supporting widely dispersed, multiple independent origins of pottery. My reanalysis with an added interaction effect in quantile regression and binomial frameworks seems supportive of this interpretation. While the interaction between time and distance does not drastically reshape the results, it provides additional nuance.

This paper can potentially reshape scholarship and pedagogy on early African ceramic traditions. I encourage the authors to continue this promising line of inquiry by formally evaluating the interaction terms in their models. Given their existing infrastructure and deeper familiarity with the data and code, they are well-positioned to conduct these analyses at full computational depth and to report whether these dynamics hold across more extensive modeling efforts.

I want to highlight that thanks to the authors' code and data, I was able to inspect their analyses in detail and provide more informed feedback. I hope it is helpful. Nonetheless, this is a feature of open science that, when conducted honestly, should facilitate further exponential growth of cumulative knowledge.

(Remarks on code availability)

See my comments within the review.

Reviewer #2

(Remarks to the Author)

(Remarks on code availability)

Reviewer #3

(Remarks to the Author)

The work I have had the pleasure to review presents a mathematical model aimed at exploring the origins of ceramic production *sensu lato* across the African continent (excluding North Africa, as this region followed a distinct historical trajectory linked to the expansion of the Mediterranean European Neolithic).

The originality of this work lies in its methodological approach, which (although previously applied in other contexts) is here employed to address the question of the invention of pottery. The study makes use of a series of statistical models based on quantile regressions, assuming an asymmetric Laplace distribution (ALD). In this regard, I consider the work to be well-formulated and consistent with the journal's editorial aims. Nevertheless, I would recommend that certain re-analyses be carried out before publication. Having offered my personal assessment, I would, with your permission, like to raise a few comments, questions, and/or reflections regarding the manuscript, in case the authors find them relevant and wish to incorporate them into the original version.

First of all, without being exhaustive, I have identified a couple of typographical errors that should be corrected. In line 71, a dot (.) following the word technology should be removed. Similarly, in line 248, the final sentence should be revised, as a punctuation mark incorrectly separates two clauses. I would now like to focus, firstly, on the methodological aspects of the study, and subsequently on the archaeological data.

With regard to the statistical modelling, the authors provide a sound justification for the use of the asymmetric Laplace distribution, although it is worth noting that this choice may pose certain challenges when working with calibrated dates—particularly due to the presence of long tails (resulting from the calibration process) and/or evidence of multimodality. In this context, have the authors considered adopting a multimodal Dirichlet approach?

The second issue, which is more archaeological in nature, concerns the information used. To what extent might the results presented be affected by the old wood effect? If the modelling were to be conducted using only short-lived dates, would the results be significantly altered?

In conclusion, as I mentioned at the beginning of my review, this work is worthy of publication, and I simply wish the authors a swift publication process, as well as the opportunity to read the final version of this manuscript.

(Remarks on code availability)

T

Reviewer #4

(Remarks to the Author)

I am very pleased to see new research examining the origins of ceramic production in Africa. I cannot provide commentary on the radiocarbon modeling as this is not my area of expertise, but I can provide general commentary on archaeological context and histories of ceramic production. Overall I think the manuscript provides a valuable contribution to the literature, particularly in its focus on improving chronological frameworks for AHP fisher/forager settlement patterning and technological innovation. I encourage its publication with revisions suggested below.

I think a reader unfamiliar with the African literature may wonder why there's very little discussion about whether closer comparative examination of ceramic production techniques could provide an answer to the single vs. multiple origin(s) question for early African pottery. It might be worth mentioning how little we know about the ceramic industry at Ounjougou, which is represented by only three sherds, found in secondary context. So, the central question here really seems to be whether the three Ounjougou sherds are simply too old to be related to other ceramics found elsewhere in northern/eastern Africa. I trust the reported TAQ and the new model results presented here, but I suspect I'm not alone in wishing we had either direct dates on organic material in those sherds or direct dates on materials found in close association with them.

The discussion is very long with very long paragraphs – it's hard to follow the argumentation here. I would use sub-headings if possible. Lines 307-333 – There is quite a bit happening in this lengthy paragraph. I would perhaps break this up into two, with one paragraph focusing on ceramic use and one on relationships between mobility, ceramic production, and ecological changes during the AHP.

In terms of ceramic use, I think this section reaches a bit far in terms of speculation. I don't believe we really know, for example, if the earliest ceramics in Africa were used for storage or not. I would limit speculation, and try to tie this discussion more directly to your results about multi-regional origins for ceramic production/use in Africa. For example, I would be tempted to argue that if ceramics were independently invented in more than one part of Africa, in different ecological/social contexts, we should shift focus to understanding ceramic use in those local contexts. Line 335-337: "The functional evolution of pottery, from detoxifying plant resources to storing vegetal and animal products, highlights its critical role in early subsistence systems" – given that the paper provides no new data about ceramic use or the functional evolution of pottery, I would delete this sentence from the conclusion. We can assume that the earliest African pots were useful to the people who made them (in ways almost certainly related to food), but in my opinion without direct residue analyses we can say very little about exactly what those uses may have been, and what they might have had to do with generalized shifts towards wetter/more humid environmental conditions.

In terms of the mobility section, I'm also not convinced by the argument (derived from 1980s-era ceramic ecology) for the necessity of "settlement stability" or "semi-residential settlement strategies" – what do these terms mean, in practice? There are examples of early ceramic production (e.g. Wu et al 2012's work in China) where problems with decreased resource availability may have been mitigated by highly mobile communities making pots. It's certainly plausible that fisher/foragers in northern Africa spent more residential time around lakes and rivers during the AHP than previously, and that this may have made ceramic production easier somehow, but I wouldn't imply that sedentism is necessarily required for ceramic production. The question for Africa is, I think, how we measure "mobility" (which can be practiced in a vast array of forms) amongst the fisher/forager communities in question, without using ceramics as proxy evidence for some form of sedentism.

The paragraph beginning "the cultural trajectories of hunter-gatherer-fishers" (Line 289) is a bit vague about what results from genetic, linguistic, and archaeological data might be telling us about population dynamics and shifts to wet/humid environmental conditions. Could the authors be a bit more explicit here, in terms of what their models are suggesting, overall, about those dynamics and their relationships to ceramic technologies?

Minor issues:

- Line 85-86: "However, they require systematic testing, especially given the scarcity of recent research due to the inherent challenges of fieldwork and the ephemeral nature of available data" – I'm not sure what this sentence means to say. Recent research hasn't happened because fieldwork is challenging? Do you mean in particular parts of northern Africa? I'm also not sure if available data are ephemeral (they're not short-lived or disappearing) – the available data are just limited. I would leave this as "However, they require systematic testing."

- Line 89: citation should be (15-16, 29) or (15, 16, 29) – I'm not sure what the correct house style is, but if 29 comes before 15-16 it appears that 15-16 are referring to page numbers. Those citations should be italicized.

- Line 163-165 "This supports the growing consensus in African archaeology that innovations such as pottery did not diffuse from a singular core area but rather emerged from multiple interconnected areas of innovation (8, 9)." I'm not sure that one author writing in the 80s and 90s is citable as evidence for "growing consensus" in African archaeology.

- Line 184 - "The concurrent development of ceramics and lithic tools across multiple regions" – at first I read this to mean that ceramics were developed concurrently across multiple regions, along with lithics. That would be incorrect. I would reword to make it more clear that you're talking about how ceramic and lithic technologies appear to change in-step with each other within specific regions (I think?), and that this same process happened in multiple regions (albeit at different times)? I would clarify this section.

- Line 190 – Change form to from

- Line 248 – typo at beginning of Raw materials

- Line 273 – typo on adopted

- Line 274 – aseffective should be two words

- Lines 337-338 – “Earlier linear diffusionist models” – it’s not made clear in the paper that earlier scholars posited linear diffusionist models for African ceramics. If so, I would think that that literature needs to be referenced in the introduction.

- Need complete citation for Huysecom 2020.

(Remarks on code availability)

Version 1:

Reviewer comments:

Reviewer #1

(Remarks to the Author)

The authors’ re-analyses now provide a decisive answer to the single versus multi-origin question, and the added interaction term strengthens the paper further. I am glad my suggestions were helpful. Their choice to test this effect only in the binomial models was exactly right, and the strong support for Model 7 speaks volumes, especially compared to the lack of support for single-origin models.

By including the interaction term, the authors show that a weak main effect of space can still reflect meaningful dynamics when considering time, which directly addresses a concern I raised in my review.

I realize that my original suggestion to include a time \times distance interaction in the quantile regression did not fully explain its rationale and potential implementation. As noted in my report, the idea came from thinking of diffusion as a self-reinforcing cultural process: “as time progresses, initial adopters become secondary transmitters.” In this framework, each new adopter acts as a transmission hub, effectively shortening cultural distances and accelerating the spread of innovation. The authors’ updated binomial interaction model captured this dynamic clearly and allowed for further meaningful inferences about the changing rate of spread.

If future work aims to explore these dynamics further, especially why diffusion may accelerate in some regions but not others, two modeling approaches could be useful. One is a latent variable approach, where diffusion momentum is treated as an unobserved, emergent process influenced by spatial structure and early CalDates, with later CalDates seen as outcomes of that momentum. Another approach is an autoregressive framework (AR, ARMA, or ARIMA), where earlier adoption dates directly affect later ones, and short-term surges in adoption are captured as part of the moving average. These are but two approaches that may be conducted under a Bayesian framework and help clarify how time can be meaningfully represented on both sides of the equation.

These methods align with the authors’ updated interpretation: as transmission networks grow stronger, cultural distance effectively shrinks.

That said, the current study stands strongly on its own, and there’s no need to reframe what is already a compelling and rigorous piece of work. I recommend this manuscript for publication as it is, and I look forward to seeing it published in Nature Communications.

(Remarks on code availability)

Reviewer #2

(Remarks to the Author)

(Remarks on code availability)

Reviewer #3

(Remarks to the Author)

The revised version of the manuscript addresses the issues raised in the initial review by the reviewers. Furthermore, in my case the justification for not employing the Dirichlet approach is appropriately articulated, as in the reasoning concerning the use of short-lived samples.

In my view, this work is now ready for acceptance.

(Remarks on code availability)

Reviewer #4

(Remarks to the Author)

Thank you for the opportunity to review the revised version of this manuscript. The authors' revisions have improved the paper and I am satisfied that all of the concerns I raised in my initial review have now been addressed. The new discussion emphasizes, for me, an acute need for additional research on contexts of early ceramic production across the African continent. My hope is that scholarship on early African ceramic production will be globally recognized in debates about forager mobility, foodways, and the invention(s) of ceramic technologies. This paper is a methodologically-important contribution to that literature and I strongly encourage its publication in Nature Communications.

(Remarks on code availability)

Rocco Rotunno
McDonald Institute for Archaeological Research, University of Cambridge, UK
West Building, McDonald Institute for Archaeological Research, Downing Street CB23ER
rotunno.rocco@gmail.com rr695@cam.ac.uk

Point-by-Point Response to Reviewer Comments for Manuscript NCOMMS-25-15778 “Bayesian analyses of radiocarbon dates suggest multiple origins of ceramic technology in Early Holocene Africa”

Please find the point-by-point responses attached below to provide a detailed account of our response to the reviewers' comments. We have included the original reviewers' comments and highlighted our responses in **green**.

REVIEWER COMMENTS

Reviewer #1 (Remarks to the Author):

Review of NCOMMS-25-15778 - Bayesian analyses of radiocarbon dates suggest multiple origins of ceramic technology in Early Holocene Africa

Thanks for allowing me to review this manuscript on the origins and diffusion of ceramic technology in Africa. As I will outline below, I really like this study. It addresses an important question in prehistory, provides a logical and innovative methodological approach to the question, and thus meets all criteria for publication in Nature Communications. I provide comments below that do not undermine the authors' arguments. Rather, my questions and suggestions regard the nature of the cultural diffusion mechanism assumed by the statistical models. Mainly, the models' ability to detect a broader spectrum of spatiotemporal diffusion signals. I believe that the authors will easily address my questions and comments. Consequently, I highly recommend this manuscript for publication in this journal.

Contextual Importance of the Study

For context, I would like to highlight that the textbook I use for my introductory archaeology course, *The Human Past: World Prehistory and the Development of Human Societies* (July 2024, 5th edition, Thames & Hudson), presents the spread of ceramics in Africa as a seemingly diffusion process from a single origin. Specifically, it states:

"From an open-air site in Mali called Ounjougou, at the southern edge of the present Sahara, comes the earliest pottery, which dates from before 9400 BCE. This is the oldest pottery found in Africa and predates that from Southwest Asia... From here, at the end of the tenth century to the beginning of the ninth century BCE, ceramics spread into other parts of Africa as the desert zone became increasingly greener." (p. 327)

To me, this framing reflects a textbook single-origin diffusionist model. It posits pottery as emerging from a single center and radiating outward in response to environmental changes.

The study is particularly significant because it directly tests this single-origin diffusion hypothesis using advanced Bayesian modeling. The study's findings, which favor multiple independent origins of pottery in Africa, challenge the single-origin diffusion model taught in classrooms (at least by the textbook I use in mine). Consequently, this paper can potentially reshape how the evolution of early African technology (e.g., ceramic traditions) is taught to future archaeologists and students more broadly.

To do this, the study combines Bayesian quantile regression and Bayesian binomial regression, along with local spatial autocorrelation analyses (I know these as "LISAs" following Anselin's work). The quantile regression framework focuses on estimating the early temporal quantile of the dataset (5th percentile), effectively isolating the earliest archaeological occurrences of pottery for analysis. One positive point to note about this approach, which others may miss, is that, like OLS, quantile regression uses all data points to estimate effects at a given quantile. Even though the study targets a specific quantile, the whole dataset contributes to the estimation. This is especially helpful in the distribution's tails, where individual quantile points are sparse. The method thus "borrows strength" in areas with potentially little data, leveraging information from the overall data structure. This strengthens the signal detection ability of their model (s).

The authors then supplement this with binomial models to estimate the probability of pottery presence across space and time. These models allow them to compare competing hypotheses of ceramic spread based on proposed origin locations and temporal dynamics.

Their results show that none of the tested centers acted as a sole origin of pottery spread, as might be expected under a classical single-origin diffusion model. Instead, the best-performing models consistently favored scenarios involving three widely geographically dispersed innovation centers. These results suggest independent emergence of ceramic technology across the continent. Their quantile regression and binomial modeling also indicate positive time effects but surprisingly small distance effects. The latter, to the extent of being zero. The probability of pottery presence does not covary with increasing distance from the proposed origins as strongly as expected from a single-origin diffusion model.

Thus, they reject the hypothesis of a single origin for the spread of pottery in Africa. These results challenge assumptions of a simple diffusion process and highlight the complexity of early African ceramic traditions.

This is an innovative, well-conducted study, and the authors should be commended for their work. First, the Bayesian approach is novel. The quantile regression ensures that only the earliest subset of radiocarbon dates is evaluated for the origins of pottery amongst themselves. Then, they use binomial Bayesian regression models to supplement their analysis with the larger sample.

I have a few suggestions regarding the structure of their models. I fully acknowledge that my thinking may be imperfect. Therefore, I will outline my reasoning openly below.

The authors are welcome to correct my reasoning if they think I am wrong. If they do so, this would naturally call my resulting suggestions into question, and they should act accordingly.

Essentially, the study's models are limited to main effects only. The covariates are standardized space (distance) and time (C14). The key is that each variable is modeled

assuming its effect does not vary across the range of the other variable. In particular, the influence of space is presumed constant across time values, and vice versa, rather than allowing for the possibility that their relationship might change over time or across spatial gradients. I strongly encourage the authors to incorporate an interaction term in their models. From the perspective of the diffusion process itself, there is good reason to expect that the rate of spatial spread does not remain constant over time. To clarify my reasoning, let me first explain how I conceptualize diffusion and why I recommend exploring this interaction effect.

I apologize if what follows seems overly detailed and lacking in citations. The authors are eminent figures in quantitatively modeling cultural transmission and ceramics, particularly in archaeology, and I deeply admire their work. Given their familiarity with diffusion models such as Wright-Fisher and MacArthur-Wilson, I will avoid belaboring well-known points but will lay out some basic thoughts for context.

My intention here is simply to explain my rationale. I believe this is a relatively straightforward and worthwhile extension that could benefit the study.

Expectations for Diffusion Data and Statistical Modeling

As given, the central pattern in diffusion processes is progressive expansion across space and time, whether of technologies like pottery, cultural innovations, or diseases. Diffusion typically begins at a localized origin point and radiates outward as time advances. In an additive model, as the study currently presents, the effect of space is treated as constant over time. This means that the spatial coefficient estimate reflects only an average influence across the full temporal range of the data. From the first principles of transmission processes, this spatial effect may appear small early on because adoption is mainly localized and the overall rate of spread is slow. However, as time progresses, initial adopters become secondary transmitters. As populations grow and interact, the influence of space amplifies, accelerating the spread across broader regions.

An additive model cannot capture this dynamic compounding effect, because it assumes spatial and temporal influences operate independently. In such cases, the estimated effect of space might appear small at first glance, which risks leading to misinterpretation. By contrast, including an interaction term between space and time allows the model to represent and test how the effect of space changes over time (and vice versa). The interaction term effectively increases the effect of space as the network of transmitters grows. Without this interaction, a model risks underestimating and failing to test the role of space. It might misinterpret a seemingly minor spatial effect as negligible, when in fact it can become a key driver of accelerating diffusion.

In this context, even modest-sized interaction estimates can exert substantial influence, particularly in models with standardized predictors and logit transformations. In these models, small shifts in parameter estimates can dramatically alter predicted probabilities.

Retooling the Bayesian Quantile Regression Models

Thanks to the code and data provided, I restructured the original Bayesian quantile regression framework to explore this further. Specifically, I included an interaction term between time and distance in their nimble quantile model code (adding: `beta_interaction * distance[i] * time[i]`; and modeling it as `beta_interaction ~ dnorm(0, sd = 1)`). This addition enables a direct test of whether the effect of distance on pottery presence changes over time.

After running the modified structure, using their "e" model, I examined each parameter's 95% Highest Posterior Density Interval (HPDI). The intercept (alpha) remained around 11,000 BP, as expected. The time and distance coefficients spanned negative to positive ranges, reflecting considerable uncertainty. Most importantly, the interaction term's 95% HPDI ranged from approximately -0.40 to $+0.94$. Although this interval includes zero, approximately 75% of the posterior distribution lies above zero. While not conclusive, this posterior skew provides moderate Bayesian support for a positive interaction effect. In other words, while distance initially suppresses the probability of pottery presence, this inhibitory effect weakens over time. This pattern aligns with expectations from diffusion theory.

I acknowledge that my reanalysis was run with fewer MCMC iterations than the original authors used. For practical reasons, I ran only a few thousand iterations compared to their 100,000 (for the quantreg) and 500,000 (for the binomial). Given that the authors are deeply familiar with their own framework and likely have optimized workflows, I strongly recommend that they run dedicated interaction models themselves to explore this hypothesis fully. They are undoubtedly better positioned to execute these models with the rigor and convergence diagnostics necessary to draw robust conclusions.

Model "e," even in my cursory evaluation, may already reveal a more nuanced spatiotemporal dynamic. Over time, distance seems to become less of a barrier to the spread of pottery. This result aligns with the core mechanisms of cultural diffusion. With a more thorough analysis, the authors can examine this pattern more deeply and evaluate its robustness.

Re-estimating the Binomial Diffusion Models with Interaction

I also re-ran the original binomial models to test space-time dynamics further. I introduced an interaction term, $\text{beta3} (\text{beta3} * \text{scaled.theta}[i] * \text{distance}[i]; \text{modeled as } \text{beta3} \sim T(\text{dnorm}(\text{mean}=0, \text{sd}=0.5), 0, \text{Inf}))$, between time and distance. This modification explicitly tests whether the effect of distance changes over time.

Comparing models using WAIC, I observed the following:

- Model m5 (e+o origins) showed a slight WAIC improvement (-1.22), suggesting the interaction modestly improves model fit.
- Model m7 (e+a+o origins) showed an increased WAIC ($+14.79$), indicating that the interaction does not benefit the fit and may overcomplicate the model.
- Other models showed minor changes, generally within the range of noise. But again, this may be due to my relatively low number of MCMC.

As noted earlier, my reruns were performed at lower iteration counts than the original authors' thorough MCMC processes. While these preliminary results suggest that the interaction term improves some models, especially those with fewer origins, I do not believe they radically change the overall model landscape. Nevertheless, the authors' indication that the spatial effect is nil merits further testing, particularly the interaction term. As I have said above, given their familiarity with the data and modeling framework, they could better investigate the role of interaction terms. Running longer chains and complete diagnostics could provide a fuller understanding.

My limited modeling of the interaction term seems to have improved models with fewer origins, like model m5, but had limited impact across the broader model set. This pattern suggests that while time and space are not wholly independent, the strength of their

interaction varies depending on the assumed number and placement of origins.

To clarify, my exercise cannot replace my recommendation that they perform a more thorough examination of my preliminary analysis. I would argue that what I did is better than providing a verbal model or thought experiment, but it remains an initial exploration. Nonetheless, I (perhaps naively) believe it highlights the potential for uncovering further dynamics within the data. Given the structure of the diffusion process and the indications from my exploratory quantile regression and binomial reanalyses, I believe this direction warrants serious attention.

Thank you for the very supportive comments and the suggestion to include an interaction term in the quantile and binomial models. We agree that introducing an additional term in the model provides an added flexibility, allowing the effect of time (or space) to vary as a function of space (or time). The reviewer suggests introducing this aspect to both the quantile and binomial models, but we have decided to do so only for the latter for two reasons. Firstly, it is unclear to us how time could be simultaneously treated as a response and a predictor in the quantile regression. Secondly, given the objective of the quantile regression was to simply demonstrate the absence of a relationship between distance and time, we considered that adding complexity to the model was unwarranted, particularly since the binomial model was chosen to address our core questions.

We thus implemented a revised version of the model, although using a different prior (a full rather than a half Gaussian to allow effects in both directions) and, of course, running a sufficient number of iterations to reach convergence. The results were somewhat different from the reviewer's analyses (although this was expected due to small number of runs carried out by them), and has slightly modified our conclusions, providing a stronger support for model 7, as well as offering some interesting insights on the interaction between time and space on the rate of adoption of ceramic technology.

We have updated all relevant portions of text, table, and figures to reflect this, as well as added a small discussion on the interpretation of the interaction term. Given our results have now suggested less support for model m5, we decided to move the original figure 2 to the SI and update figure 4 accordingly.

Code Comments and Reproducibility Notes

While revising the code, I encountered a few minor issues:

1. The `here()` package requires proper project root recognition. For reproducibility, it might help to set the working directory explicitly or use an RStudio Project to manage paths. Maybe because I am old school (it hurts a little to say that!), I replaced `load(here('data','input_binom.RData'))` with `load("../data/input_binom.RData")`.

We added an RProj file to allow proper project root recognition.

2. The function `select()` was ambiguous in my environment due to conflicts with other loaded packages. I imagine others may have a similar problem. Explicitly using `dplyr::select()` resolved this.

We updated the script with an explicit call to `dplyr`

3. I spent considerable time understanding the loop structures handling stratigraphic constraints. Some additional inline comments in the future would enhance clarity and ease replication.

We added comments on the relevant lines of our script.

These adjustments improve robustness and reproducibility across environments.

Summary

In conclusion, the paper under evaluation offers a methodologically rigorous and empirically grounded challenge to single-origin diffusion hypotheses of African ceramics. By employing Bayesian frameworks, the authors provide evidence supporting widely dispersed, multiple independent origins of pottery. My reanalysis with an added interaction effect in quantile regression and binomial frameworks seems supportive of this interpretation. While the interaction between time and distance does not drastically reshape the results, it provides additional nuance.

This paper can potentially reshape scholarship and pedagogy on early African ceramic traditions. I encourage the authors to continue this promising line of inquiry by formally evaluating the interaction terms in their models. Given their existing infrastructure and deeper familiarity with the data and code, they are well-positioned to conduct these analyses at full computational depth and to report whether these dynamics hold across more extensive modeling efforts.

I want to highlight that thanks to the authors' code and data, I was able to inspect their analyses in detail and provide more informed feedback. I hope it is helpful. Nonetheless, this is a feature of open science that, when conducted honestly, should facilitate further exponential growth of cumulative knowledge.

Thank you!

Reviewer #1 (Remarks on code availability):

See my comments within the review.

Reviewer #2 (Remarks to the Author):

Reviewer #3 (Remarks to the Author):

The work I have had the pleasure to review presents a mathematical model aimed at exploring the origins of ceramic production sensu lato across the African continent (excluding North Africa, as this region followed a distinct historical trajectory linked to the expansion of the Mediterranean European Neolithic).

The originality of this work lies in its methodological approach, which (although previously applied in other contexts) is here employed to address the question of the invention of pottery. The study makes use of a series of statistical models based on quantile regressions, assuming an asymmetric Laplace distribution (ALD). In this regard, I consider the work to be well-formulated and consistent with the journal's editorial aims. Nevertheless, I would recommend that certain re-analyses be carried out before publication. Having offered my personal assessment, I would, with your permission, like to raise a few comments, questions, and/or reflections regarding the manuscript, in case the authors find them relevant and wish to incorporate them into the original version.

Comment 1:

First of all, without being exhaustive, I have identified a couple of typographical errors that should be corrected. In line 71, a dot (.) following the word technology should be removed. Similarly, in line 248, the final sentence should be revised, as a punctuation mark incorrectly separates two clauses. I would now like to focus, firstly, on the methodological aspects of the study, and subsequently on the archaeological data.

Thank you for the comment, we revised and amended the text for typos, including the examples raised by the reviewer.

Comment 2:

With regard to the statistical modelling, the authors provide a sound justification for the use of the asymmetric Laplace distribution, although it is worth noting that this choice may pose certain challenges when working with calibrated dates—particularly due to the presence of long tails (resulting from the calibration process) and/or evidence of multimodality. In this context, have the authors considered adopting a multimodal Dirichlet approach?

We decided to keep the quantile regression for two reasons. Firstly the approach has been successfully adopted (and tested via simulated datasets) on previous applications (e.g. DOI: 10.1126/sciadv.adc9171; DOI:10.1057/s41599-021-00717-w; DOI: 10.1371/journal.pone.0137024). Secondly, given our focus on extreme values/quantile we consider the effect of the long tail comparatively negligible. Thirdly, given the purpose of the quantile regression is to primarily highlight (with a different statistical approach) the lack of empirical evidence supporting single origin models, we believe our approach is adequate in this case.

Comment 3:

The second issue, which is more archaeological in nature, concerns the information used. To what extent might the results presented be affected by the old wood effect? If the modelling were to be conducted using only short-lived dates, would the results be significantly altered?

Thank you for raising this point. The potential impact of the old wood effect is acknowledged, particularly for dates derived from charcoal samples where the species is unknown. However, the dataset also includes a proportion of dates from short-lived materials, which altogether do reduce flaws in the model. While re-running the models using only short-lived dates would reduce the sample size and geographic coverage, we agree that further targeted short-lived dating would be valuable to refine these models.

In conclusion, as I mentioned at the beginning of my review, this work is worthy of publication, and I simply wish the authors a swift publication process, as well as the opportunity to read the final version of this manuscript.

Reviewer #3 (Remarks on code availability):

T

Reviewer #4 (Remarks to the Author):

I am very pleased to see new research examining the origins of ceramic production in Africa. I cannot provide commentary on the radiocarbon modeling as this is not my area of expertise, but I can provide general commentary on archaeological context and histories of ceramic production. Overall I think the manuscript provides a valuable contribution to the literature, particularly in its focus on improving chronological frameworks for AHP fisher/forager settlement patterning and technological innovation. I encourage its publication with revisions suggested below.

Comment 1:

I think a reader unfamiliar with the African literature may wonder why there's very little discussion about whether closer comparative examination of ceramic production techniques could provide an answer to the single vs. multiple origin(s) question for early African pottery. It might be worth mentioning how little we know about the ceramic industry at Ounjougou, which is represented by only three sherds, found in secondary context. So, the central question here really seems to be whether the three Ounjougou sherds are simply too old to be related to other ceramics found elsewhere in northern/eastern Africa. I trust the reported TAQ and the new model results presented here, but I suspect I'm not alone in wishing we had either direct dates on organic material in those sherds or direct dates on materials found in close association with them.

Thank you for raising this important point. We agree that ceramic production techniques could help clarify the question of single vs. multiple origins, but such comparative data are still limited, especially for the earliest sites. In the case of Ounjougou, the evidence is indeed minimal—just three sherds in secondary context—and their technological features remain poorly known. While we trust the TAQ and the new chronological model, we acknowledge the limitations and agree that more direct dating, ideally on residues or associated materials, would greatly strengthen the case. We have tried to present the Ounjougou evidence cautiously in light of these constraints and as already highlighted in the manuscript here we simply attempt to present a valuable and robust model to test some of the hypotheses with all the inherent limitations.

Comment 2:

The discussion is very long with very long paragraphs – it's hard to follow the argumentation here. I would use sub-headings if possible. Lines 307-333 – There is quite a bit happening in this lengthy paragraph. I would perhaps break this up into two, with one paragraph focusing

on ceramic use and one on relationships between mobility, ceramic production, and ecological changes during the AHP.

We divided the paragraph as suggested and also modified the text accordingly. We rewrote the whole section, see Comment 3 below.

Comment 3:

In terms of ceramic use, I think this section reaches a bit far in terms of speculation. I don't believe we really know, for example, if the earliest ceramics in Africa were used for storage or not. I would limit speculation, and try to tie this discussion more directly to your results about multi-regional origins for ceramic production/use in Africa. For example, I would be tempted to argue that if ceramics were independently invented in more than one part of Africa, in different ecological/social contexts, we should shift focus to understanding ceramic use in those local contexts. Line 335-337: "The functional evolution of pottery, from detoxifying plant resources to storing vegetal and animal products, highlights its critical role in early subsistence systems" – given that the paper provides no new data about ceramic use or the functional evolution of pottery, I would delete this sentence from the conclusion. We can assume that the earliest African pots were useful to the people who made them (in ways almost certainly related to food), but in my opinion without direct residue analyses we can say very little about exactly what those uses may have been, and what they might have had to do with generalized shifts towards wetter/more humid environmental conditions.

Thank you for this useful comment. We addressed the various points raised and agree that there is sparse evidence on pottery use, especially related to the earliest pottery-bearing contexts, though, as evidenced in some of the cited references, some hints can be grasped from very specific studies. Nonetheless, we modified the entire paragraph, limiting speculation about use contexts and emphasising the need for a more localised eco-cultural approach.

We also modified the conclusions by deleting the part related to use and function as follows:

~~"In conclusion, our statistical analyses suggest that rather than originating from a single innovation centre, multiple regions contributed to the broader adoption of ceramics across the continent. The functional evolution of pottery, from detoxifying plant resources to storing vegetal and animal products, highlights its critical role in early subsistence systems. These findings challenge earlier linear diffusionist models and underscore the necessity of ongoing archaeological research to brighten the intricate cultural and technological processes that shaped prehistoric African societies, underpinning the socio-economic developments of Early Holocene Africa."~~

Comment 4:

In terms of the mobility section, I'm also not convinced by the argument (derived from 1980s-era ceramic ecology) for the necessity of "settlement stability" or "semi-residential settlement strategies" – what do these terms mean, in practice? There are examples of early ceramic production (e.g. Wu et al 2012's work in China) where problems with decreased resource availability may have been mitigated by highly mobile communities making pots. It's certainly plausible that fisher/foragers in northern Africa spent more residential time around lakes and rivers during the AHP than previously, and that this may have made ceramic production easier somehow, but I wouldn't imply that sedentism is necessarily required for ceramic production. The question for Africa is, I think, how we measure "mobility" (which can be

practiced in a vast array of forms) amongst the fisher/forager communities in question, without using ceramics as proxy evidence for some form of sedentism.

We appreciate this insightful comment and fully agree that ceramic production does not require sedentism, and that mobility takes many forms. We have revised the text to clarify that we do not assume a deterministic relationship between reduced mobility and pottery-making. Rather, we refer to logistically organised settlement strategies as one possible framework—emerging under African Humid Period conditions—that may have facilitated repeated access to raw materials. Importantly, we do not use ceramics as a proxy for sedentism, but situate their appearance within broader socio-ecological shifts that also influenced mobility practices.

Comment 5:

The paragraph beginning “the cultural trajectories of hunter-gatherer-fishers” (Line 289) is a bit vague about what results from genetic, linguistic, and archaeological data might be telling us about population dynamics and shifts to wet/humid environmental conditions. Could the authors be a bit more explicit here, in terms of what their models are suggesting, overall, about those dynamics and their relationships to ceramic technologies?

We thank the reviewer for this helpful suggestion. We have revised the paragraph to clarify how genetic, linguistic, and archaeological data converge to support models of regional demographic continuity, cultural transmission, and localized innovation in response to shifting environmental conditions during the African Humid Period.

Minor issues:

- Line 85-86: “However, they require systematic testing, especially given the scarcity of recent research due to the inherent challenges of fieldwork and the ephemeral nature of available data” – I’m not sure what this sentence means to say. Recent research hasn’t happened because fieldwork is challenging? Do you mean in particular parts of northern Africa? I’m also not sure if available data are ephemeral (they’re not short-lived or disappearing) – the available data are just limited. I would leave this as “However, they require systematic testing.”

We modified the text in lines 85-86 as suggested.

- Line 89: citation should be (15-16, 29) or (15, 16, 29) – I’m not sure what the correct house style is, but if 29 comes before 15-16 it appears that 15-16 are referring to page numbers. Those citations should be italicized.

Corrected.

- Line 163-165 “This supports the growing consensus in African archaeology that innovations such as pottery did not diffuse from a singular core area but rather emerged from multiple interconnected areas of innovation (8, 9).” I’m not sure that one author writing in the 80s and 90s is citable as evidence for “growing consensus” in African archaeology.

Thank you for this helpful observation. We agree that the phrasing “growing consensus” overstates the support currently represented by the cited references. We have revised the

sentence to more accurately reflect the available evidence and now state: *“This supports the view that innovations such as pottery did not originate from a single core area but rather emerged in multiple, potentially interconnected regions of innovation (8, 9).”*

- Line 184 - “The concurrent development of ceramics and lithic tools across multiple regions” – at first I read this to mean that ceramics were developed concurrently across multiple regions, along with lithics. That would be incorrect. I would reword to make it more clear that you’re talking about how ceramic and lithic technologies appear to change in-step with each other within specific regions (I think?), and that this same process happened in multiple regions (albeit at different times)? I would clarify this section.

Thank you for the useful comment. We modified the text to address our reasoning more clearly, in the following way: [...] *↯ suggests that the adoption of pottery occurred within a wider context of technological renewal. Rather than emerging in isolation, early ceramics appear to have developed alongside changes in lithic production within distinct cultural settings. The recurrence of this association in various regions points to a shared dynamic in which technological shifts were shaped by local adaptations to environmental change and population reorganisation.*

- Line 190 – Change form to from

We modified the text as suggested.

- Line 248 – typo at beginning of Raw materials

We modified accordingly

- Line 273 – typo on adopted

Amended

- Line 274 – aseffective should be two words

Amended

- Lines 337-338 – “Earlier linear diffusionist models” – it’s not made clear in the paper that earlier scholars posited linear diffusionist models for African ceramics. If so, I would think that that literature needs to be referenced in the introduction.

We modified the conclusions, see comment 3 above.

- Need complete citation for Huysecom 2020.

Amended

Rocco Rotunno
McDonald Institute for Archaeological Research, University of Cambridge, UK
West Building, McDonald Institute for Archaeological Research, Downing Street CB23ER
rotunno.rocco@gmail.com rr695@cam.ac.uk

Response to Reviewer Comments for Manuscript NCOMMS-25-15778 “Bayesian analyses of radiocarbon dates suggest multiple origins of ceramic technology in Early Holocene Africa”

Dear Editor,

We are pleased to submit the revised version of our manuscript titled "Bayesian analyses of radiocarbon dates suggest multiple origins of ceramic technology in Early Holocene Africa" (Manuscript ID: NCOMMS-25-15778) for your consideration. We are grateful for the thoughtful and constructive comments provided by the reviewers.

Please find in **green** our responses to the reviewers' comments

REVIEWER COMMENTS

Reviewer #1 (Remarks to the Author):

The authors' re-analyses now provide a decisive answer to the single versus multi-origin question, and the added interaction term strengthens the paper further. I am glad my suggestions were helpful. Their choice to test this effect only in the binomial models was exactly right, and the strong support for Model 7 speaks volumes, especially compared to the lack of support for single-origin models.

By including the interaction term, the authors show that a weak main effect of space can still reflect meaningful dynamics when considering time, which directly addresses a concern I raised in my review.

I realize that my original suggestion to include a time \times distance interaction in the quantile regression did not fully explain its rationale and potential implementation. As noted in my report, the idea came from thinking of diffusion as a self-reinforcing cultural process: "as time progresses, initial adopters become secondary transmitters." In this framework, each new adopter acts as a transmission hub, effectively shortening cultural distances and accelerating the spread of innovation. The authors' updated binomial interaction model captured this dynamic clearly and allowed for further meaningful inferences about the changing rate of spread.

If future work aims to explore these dynamics further, especially why diffusion may accelerate in some regions but not others, two modeling approaches could be useful. One is

a latent variable approach, where diffusion momentum is treated as an unobserved, emergent process influenced by spatial structure and early CalDates, with later CalDates seen as outcomes of that momentum. Another approach is an autoregressive framework (AR, ARMA, or ARIMA), where earlier adoption dates directly affect later ones, and short-term surges in adoption are captured as part of the moving average. These are but two approaches that may be conducted under a Bayesian framework and help clarify how time can be meaningfully represented on both sides of the equation.

These methods align with the authors' updated interpretation: as transmission networks grow stronger, cultural distance effectively shrinks.

That said, the current study stands strongly on its own, and there's no need to reframe what is already a compelling and rigorous piece of work. I recommend this manuscript for publication as it is, and I look forward to seeing it published in Nature Communications.

We sincerely thank the reviewer for their detailed and thoughtful feedback. We are grateful that our re-analyses addressed their concerns and that the inclusion of the interaction term strengthened the study as they suggested. Their insights into the rationale for the time \times distance interaction and the broader dynamics of cultural diffusion were especially valuable in shaping our revisions. We also greatly appreciate the suggestions regarding latent variable and autoregressive approaches, which provide promising avenues for future work. We are delighted that the reviewer considers the manuscript compelling in its current form and thank them for their recommendation for publication.

Reviewer #2 (Remarks to the Author):

Thank you.

Reviewer #3 (Remarks to the Author):

The revised version of the manuscript addresses the issues raised in the initial review by the reviewers. Furthermore, in my case the justification for not employing the Dirichlet approach is appropriately articulated, as in the reasoning concerning the use of short-lived samples. In my view, this work is now ready for acceptance.

We thank the reviewer for their positive assessment of our revised manuscript and for acknowledging our clarifications regarding the Dirichlet approach and the use of short-lived samples. We are grateful for their recommendation of acceptance.

Reviewer #4 (Remarks to the Author):

Thank you for the opportunity to review the revised version of this manuscript. The authors' revisions have improved the paper and I am satisfied that all of the concerns I raised in my initial review have now been addressed. The new discussion emphasizes, for me, an acute need for additional research on contexts of early ceramic production across the African continent. My hope is that scholarship on early African ceramic production will be globally recognized in debates about forager mobility, foodways, and the invention(s) of ceramic technologies. This paper is a methodologically-important contribution to that literature and I strongly encourage its publication in Nature Communications.

We sincerely thank the reviewer for their positive evaluation of our revised manuscript and for recognizing the methodological contribution of our study. We are especially grateful for their encouragement to situate early African ceramic production within wider global debates, and we share their hope that our research will contribute to advancing this discussion.